# Domain-Specific Processing Stage for Estimating Single-Trail Evoked Potential Improves CNN Performance in Detecting Error Potential

**DOI:** 10.3390/s23229049

**Published:** 2023-11-08

**Authors:** Andrea Farabbi, Luca Mainardi

**Affiliations:** Department of Electronics, Information and Bioengineering, Politecnico di Milano, 20133 Milan, Italy; andrea.farabbi@polimi.it

**Keywords:** Brain–Computer Interface, signal processing, electroencephalography, Error Potential, Single-Trial analysis, deep learning, machine learning

## Abstract

We present a novel architecture designed to enhance the detection of Error Potential (ErrP) signals during ErrP stimulation tasks. In the context of predicting ErrP presence, conventional Convolutional Neural Networks (CNNs) typically accept a raw EEG signal as input, encompassing both the information associated with the evoked potential and the background activity, which can potentially diminish predictive accuracy. Our approach involves advanced Single-Trial (ST) ErrP enhancement techniques for processing raw EEG signals in the initial stage, followed by CNNs for discerning between ErrP and NonErrP segments in the second stage. We tested different combinations of methods and CNNs. As far as ST ErrP estimation is concerned, we examined various methods encompassing subspace regularization techniques, Continuous Wavelet Transform, and ARX models. For the classification stage, we evaluated the performance of EEGNet, CNN, and a Siamese Neural Network. A comparative analysis against the method of directly applying CNNs to raw EEG signals revealed the advantages of our architecture. Leveraging subspace regularization yielded the best improvement in classification metrics, at up to 14% in balanced accuracy and 13.4% in *F1-score*.

## 1. Introduction

The development of Brain–Computer Interface (BCI) systems holds promise for a wide range of applications, from assisting individuals with disabilities to enhancing human–computer interaction in various domains [1].

In this, Electroencephalography (EEG) has played a pivotal role in BCI research, providing a non-invasive means of capturing neural activity with high temporal resolution [2]. In particular, the Error Potential (ErrP) has gained attention due to its ability to optimize BCI system performance [3]. The ErrP is an Evoked Potential (EP) that represents the neural response associated with the detection and elaboration of mistakes made by the subject itself, by another subject, or by a machine [4]. The activity is located in the medio-frontal areas of the brain, in particular in the anterior cingulate cortex; depending on the task, different ErrPs types can be distinguished. Their realization follows a stereotypical shape characterized by a negative peak occurring at 250 ms after the error, a positive peak at 320 ms, and another negative peak at 450 ms. In the frequency domain, the EEG signal recorded after the erroneous event is particularly localized in the δ (1–3 Hz) and θ (5–8 Hz) brain rhythms [5].

The ErrP holds significant value within BCI systems, serving as a critical resource for rectifying the system’s erroneous outputs and enhancing overall performance [6]. Consequently, the extraction of the ErrP waveform from the EEG signal represents a crucial step; however, achieving precise separation of the ErrP from the underlying EEG background remains a non-trivial and complex challenge [7,8].

### 1.1. ErrP Classification by NN

Recently, the problem of distinguishing between ErrP and Non-ErrP epochs has been approached using Deep Learning algorithms, particularly Convolutional Neural Networks (CNNs).

The study in [9] employed a CNN specifically fine-tuned for ErrP recognition. The architecture of this CNN comprised four convolutional blocks, rendering it a deep CNN (DNN). The network’s input consisted of EEG signals epoched in the second following the erroneous stimulus. The signal preprocessing was standard, involving common average re-referencing, band-pass filtering within the desired frequency range, and the exclusion of corrupted electrodes. Furthermore, only the Cz and FCz channels were selected as the input to the CNN, neglecting information from other channels. The results obtained from a publicly available dataset demonstrated test accuracy performance of up to 86.1%.

Lawhern et al. [10] introduced a novel CNN called EEGNet. This CNN is versatile and has been tested across different BCI paradigms (e.g., ErrP, P300, Motor Imagery). The proposed architecture is simple, comprising only two convolutional blocks for frequency and spatial filtering. The low dimension of the architecture facilitates training even with limited data, a significant advantage when working with deep learning algorithms. ErrP prediction results were presented in terms of accuracy following four-fold cross-validation, with the authors claiming performance of up to 83%. In this case, the input signals underwent minimal preprocessing restricted to band-pass filtering within the frequency band of interest (i.e., 1–40 Hz).

In [11], the authors compared the ErrP classification results achieved by using a simple CNN with two different inputs: signals recorded at electrodes Cz and FCz, and signals from all 64 channels. Additionally, they compared the proposed architecture with more traditional machine learning algorithms, i.e., Support Vector Machine, Gaussian classifiers, and Deep Belief Networks. EEG signal preprocessing included frequency filtering, artifact removal using Independent Component Analysis, whitening, and cropping. Their architecture outperformed other baseline methods in terms of validation accuracy, reaching a performance of up to 84.4%. Notably, the best performance was attained when using information from all channels as input to the CNN.

Luo et al. [12] utilized a simple and compact CNN with two convolutional blocks to identify ErrP epochs, which were subsequently used as training data for a reinforcement learning algorithm. The authors applied notch filtering at 50 Hz to reduce interference and applied band-pass filtering within the range of 0.1–30 Hz. The study reported performance in distinguishing ErrP and Non-ErrP epochs in terms of Area Under the Curve (AUC) in the test set, achieving results of up to 88%.

Lastly, we mention the work presented in [13], where the authors proposed a Siamese Neural Network for a four-class problem in predicting Motor Imagery (MI) events. Although this study did not primarily focus on ErrP, the obtained results and adopted architecture hold promise. The authors introduced a neural network scheme in which two identical CNNs were trained in order to determine whether the input of one CNN matched that of the other. The paper did not describe any specific preprocessing steps applied to the data. The results for the multi-class problem were presented as the classifier’s performance compared to random labeling of unseen trials, reaching a performance of up to 86.1% on the test set.

All of these studies share the same approach in common, involving a minimal EEG pre-processing step followed by a CNN/DNN classification stage. This methodology entrusts all domain-specific processing to the CNN/DNN stage, making the DNN tasks more difficult and potentially having detrimental impacts on the classification performance.

### 1.2. Single Trial-Estimation

Different approaches and techniques have been proposed in the literature, ranging from traditional signal averaging to more advanced spatial filtering and time–frequency analysis. The most trivial technique is the Grand Average [14], which consists of averaging the epochs of interest in order to zero out the contribution of the background EEG and emphasize the EP content. However, the main drawback of this method is that the information of the Single Trial (ST) is lost; thus, this approach cannot by applied for real-time detection of the EP of interest [15]. Moreover, the SNR of the signal obtained with this technique is highly dependent on the number of epochs used for averaging. Other methods use more sophisticated filters in order to estimate the ST-ErrP. The first technique presented for ST analysis is ARX modeling [16], a method which estimates the ErrP-ST as a filtered version of the reference ErrP waveform. However, this method finds little application in real-time procedures, as estimating a new ARX model stimulus-by-stimulus can be a computationally demanding and time-consuming procedure.

Denoho et al. [17] proposed a method for estimating ErrP-ST based on the Wavelet Continuous Transform (CWT) and the Mallat algorithm [18]. With these methods, it is possible to decompose the signal at different frequency scales, thereby obtaining detailed and approximated versions of the signal in different frequency bands. At each scale, a set of coefficients is obtained and compared to a threshold by retaining those coefficients related to ErrP and eliminating those related to the background EEG. However, the selection of an appropriate threshold is highly empirical, and the method may not be robust for all applications.

Another interesting method for EP extraction is subspace regularization [19]. Through Bayesian estimation, it is possible to extract the ST from the EEG signal. The computation of the ST epoch is fast, making it suitable for real-time purposes. In the literature, the technique has been found to work well with different types of EPs. However, the main drawback of this method is that a robust estimate of the background is required (i.e., ARX modeling for estimating the background [20]), which can introduce additional computational time.

### 1.3. Aim of This Study

In this work, we hypothesize that the potential of ST techniques for domain-specific filtering of EEG data can be beneficial in improving the classification of ErrP vs. non-ErrP epochs as performed by CNNs. To validate this hypothesis, we have developed a novel architecture that integrates a domain-specific block used for ErrP-ST estimation into the conventional EEG classification framework consisting of initial EEG preprocessing followed by CNN classification. We demonstrate that the incorporation of this new block can significantly enhance the performance of the CNN classifier.

## 2. Materials and Methods

This section describes the pre-processing performed on the EEG signal to extract the epochs of interest, the different methods used for ST-ErrP extraction, and the classification performed by different CNNs. These steps are summarized in the pipeline shown in Figure 1 are described in detail in the following sections.

### 2.1. EEG Dataset

We analyzed EEG data contained in the open access BCI dataset *BNCI Horizon 2020: Monitoring error-related potentials* [21].

This dataset consists of EEG recordings obtained during an ErrP-specific experiment performed on six subjects (mean age 27.83 ± 2.23) in two recording sessions [22].

The experimental paradigm consisted of reaching a target (i.e., a colored square) through a moving cursor. The working area involved twenty possible horizontal positions at which the cursor and the target square could be located. At each time step of 2 s each, the cursor moves a step toward the target. When the target is reached, the cursor remains in place and a new target location appears. Subjects were asked to monitor the position of the cursor without having any control over it, knowing that the objective is to reach the target. In order to elicit the ErrP, there is a 20% probability at each time step of the cursor being moved in the wrong direction.

Each recording session consisted of ten blocks of 3 min each, including approximately fifty cursor movements per block. Subjects performed two recording sessions with a gap of several weeks. For each session, the EEG signal was recorded (512 Hz sampling frequency) with 64 electrodes using a BioSemi ActiveTwo system. Electrodes were placed according to the 10–20 International System.

This data-set is largely unbalanced, being constituted by 6437 epochs, of which only 1322 include the ErrP.

### 2.2. Data Preprocessing

A preprocessing pipeline was defined to extract the segments of the signal (epochs) localized just after the presentation of the feedback in which the ErrP may be present. Raw EEG data were spatially filtered with the Common Average Reference (CAR) approach and then band-pass filtered using an FIR filter in the range $\left[1\;40\right]$ Hz (as suggested in [5]), as the ErrP can be considered a relatively slow cortical potential. Data were then downsampled at 64 Hz and divided into epochs covering the interval $\left[-1\;\;1\right]$s from the instant of cursor movement. This range was chosen in order to include the pre-stimulus window and cover the expected latency/duration of the ErrP signal [5].

In view of the real-time application of the proposed method and in order to ensure an easy and low-computation procedure, no removal of artifacts, eye blinking, or eye movement was employed. While most of the methods for ErrP detection/classification [23] have analyzed data from FCz and Cz channels only, we processed all of the channels in order to include all the information captured by the electrodes.

### 2.3. Single-Trial Estimation

The epochs were then processed using different ST estimate techniques: the subspace regularization method, ARX modeling, and Continuous Wavelet Transform, which are described in the following subsections.

#### 2.3.1. Subspace Regularization

This method was first introduced in [19] to separate the ErrP waveform from the background ECG noise by incorporating second-order statistical information, specifically, the variances of the evoked potentials. The main concept is to devise a filter that can effectively enhance the signal of interest from among the extraneous brain activity that is unrelated to the mechanisms generating the ErrP. Through this regularization approach, provided that applicable density assumptions for both the signal and noise are satisfied, the obtained solutions can be viewed as Bayesian point estimates. It has been ascertained that the signal under observation (*z*) is a combination of the signal of relevance (*s*) and interference (*v*), which is commonly referred to as spontaneous EEG. The model is
(1)z=s+v=Hθ+v,
where *H* is a matrix with columns that are *p* basis functions $\psi_{i}$ to be selected according to the specific problem; moreover,
(2)$$H=(\psi_{1},...,\psi_{p})$$
where the basis functions are Gaussian-shaped components preselected for their shape and varied in terms of their delays ($\tau_{i}$), i.e.,
(3)$$\psi_{i}=e^{-\frac{1}{2d^{2}}\left(t-\tau_{i}^{2}\right)},$$
where d=0.1 is the width parameter and the number of basis functions *p* is set to 20. The ErrP for the individual tests is determined using the following formula:(4)$$\hat{s}={\left(I+\alpha_{2}^{2}D_{d}^{T}D_{d}\right)}^{-1}H{\left(H^{T}C_{v}^{-1}H+\alpha^{2}H^{t}\left(I-K_{s}K_{s}^{T}\right)H\right)}^{-1}H^{T}C_{v}^{-1}z$$
where α and $\alpha_{2}$ are regularization terms obtained empirically (set to 0.01 and 10, respectively, in our study), D is the second difference matrix, $C_{v}$ is the estimate of the covariance matrix of the noise, and $K_{s}$ is the matrix of the first eight eigenvectors of the correlation matrix of the recorded signal z. To estimate the properties of the background, we modified the original method, which employed an ARX estimate, to simply use the EEG signal recorded in the second before the stimulus. It is reasonable to assume that the properties of the background EEG during the stimulus do not significantly change from the second before the stimulus itself. This choice was motivated by the need to reduce computational burden in the view of the online application of our method.

#### 2.3.2. ARX Modeling

The ARX model for single-sweep estimation is presented in [16]. The recorded EEG signal can be described by the following equation:(5)$$y_{i}\left(t\right)=\sum_{j=1}^{p}a_{j}y_{i}\left(t-j\right)+\sum_{k=1}^{q}b_{k}u\left(t-k-d\right)+e_{i}\left(t\right),$$
where the $a_{j}$s and $b_{k}$s are the model coefficients for the AR and exogenous parts, respectively, *p* and *q* are the model orders, and *d* is a delay lag. The model is fed by $e_{i}\left(t\right)$, e.g., a white noise process.

The AR part of Equation (Equation 5) is used to model the background EEG activity (not related to error processing), while the exogenous input u(·) is used to model the expected waveform of the ErrP, and is commonly obtained by synchronous averaging of the epochs containing ErrP. The model in Equation (Equation 5) can be rewritten in the z-transform domain, as follows:(6)$$Y_{i}\left(z\right)=\underset{ErrP}{\underset{︸}{\frac{B_{i}\left(z\right)}{A_{i}\left(z\right)}U\left(z\right)}}+\underset{Background\;EEG}{\underset{︸}{\frac{1}{A_{i}\left(z\right)}E_{i}\left(z\right)}}$$
where it emerges from the similarity with Equation (Equation 1) that the ErrP is obtained as a filtered version of the exogenous input U(z), while the EEG is modeled as a colored version of a Gaussian white noise process. After having estimated u(·) for each patient, the model can be identified by estimating the coefficients $a_{j}$ and $b_{k}$, the polynomial order *p* and *q*, and the delay *d*. To identify the coefficients, a *least square method* approach is performed to minimize the following figure of merit:(7)$$F_{i}\left(t\right)=\frac{1}{N}\sum_{j=1}^{N}{\left(y_{i}\left(t\right)-{\hat{y}}_{i}\left(t\right)\right)}^{2},$$
where *N* is the number of time samples, $y_{i}\left(t\right)$ is the signal itself, and ${\hat{y}}_{i}\left(t\right)$ is the model’s prediction.

The optimal model orders are selected as those resulting from the minimization of the Akaike Information Criterion (AIC). The model identification is repeated at each *i*-th epoch and each single evoked response is described through a different model.

#### 2.3.3. Continuous Wavelet Transform

The CWT method was presented by Ahmadi et al. [24] and is based on the Donoho ST estimate technique [17]. The wavelet transform is the inner product of a signal with dilated and translated versions of a wavelet function. For a given signal x(t) and wavelet function $\psi_{a},b\left(t\right)$, the Continuous Wavelet Transform (CWT) is defined as follows:(8)$$W_{\psi }X\left(a,b\right)=<x,\psi_{a,b}>,$$
(9)$$\psi_{a,b}={\left|a\right|}^{-\frac{1}{2}}\psi \left(\frac{t-b}{a}\right),$$
where a,b∈R are the scale and translation parameters, respectively. Through the Mallat algorithm, the details and approximation of the signal can be obtained at different scales with corresponding coefficients. The original work [24] identified a five-scale decomposition and a quadratic B-Spline wavelet as being the most suitable for EP analysis. The CWT is very redundant; without incurring any loss of information, it is more practical to define the wavelet transform only at a discrete set of scales $a_{j}=2^{j}$ and times $b_{j,k}=2^{j}k$, thereby obtaining the Dyadic Wavelet Transform (DWT).

The estimation of ErrP is obtained by removing the EEG activity through denoising, which is implemented by selecting DWT coefficients based on scale-dependent thresholding. At each scale *j*, the threshold is computed as
(10)$$T_{j}^{2}=\sigma_{j}^{2}\sqrt{2\log_{e}N},$$
where $\sigma_{j}$ is the estimate of the standard deviation of the noise and *N* is the number of wavelet coefficients. In order to incorporate information from the neighboring coefficients, the thresholding criterion is as follows:(11)$$X_{den}\left(j,k\right)=\left\{\begin{array}{cc} X\left(j,k\right) & \mathrm{if}\;X_{j,k-1}^{2}+X_{j,k}^{2}+X_{j,k+1}^{2}>T_{j}^{2}\\ 0 & \mathrm{if}\;X_{j,k-1}^{2}+X_{j,k}^{2}+X_{j,k+1}^{2}\leq T_{j}^{2} \end{array}\right.$$

Moreover, the original authors introduced a dependency between coefficients at coarser scales and those at finer scales such that a coarser coefficient is removed, then its “children” in the finer scales are removed as well. First, the coefficients are selected using the grand average of the epochs of interest as the input signal, then the same coefficients are selected in order to denoise the single epochs and emphasize the ErrP part.

### 2.4. Classification

Classification of ErrP versus non-ErrP epochs was obtained by three CNN architectures: EEGNet [10], the CNN presented by Luo et al. [12], and a Siamese Neural Network. All of these CNNs have been reported to achieve good performance for BCI purposes, especially for the detection of ErrP epochs. The models obtained with the three architectures were fitted using the *Adam* algorithm defined in [25] and while minimizing the *binary cross-entropy*. They were trained through 300 epochs with a batch size of 16 on an NVIDIA GeForce RTX 2060 GPU. To evaluate the classifier performances, two approaches were followed, namely, population-wise and subject–wise. In the former, all subject data were considered as a unique dataset, while in the latter each subject was considered singularly.

The dataset was shuffled to eliminate any bias related to the sequence of stimulus, then partitioned, with 20% of the data used as a test set and the remaining data subdivided into 80% in the training set and 20% in the validation set. Stratified five-fold cross-validation was performed on the validation set. A majority vote approach was employed to obtain the prediction label for the test set, which was chosen to ensure that it could outperform the best classifier if the classifiers made independent errors [26].

The training set was balanced to obtain an equal amount of epochs for each class. In particular, the ARX balancing technique introduced in [27] was used, as it has previously been found to be robust, especially in ErrP applications.

The three CNNs we used are described in the following subsections.

#### 2.4.1. EEGNet

EEGNet was chosen because of its proven ability to generalize across different EEG-based BCI paradigms, including the ErrP classification, and because it presents a limited number of trainable parameters. The main peculiarity of this network is that it includes both depthwise and separable convolution layers, which are usually adopted in computer vision [28] to find the optimal volume-wise filters while having a reduced number of parameters compared to the classical convolution filter approach.

The architecture can be subdivided into three blocks; the last is a fully connected layer, while the others include dropout and batch normalization layers to prevent overfitting along with pooling layers to reduce the size of the data.

#### 2.4.2. L–CNN

The L-CNN architecture has two convolutional layers, one pooling layer, and one dense layer. The first convolutional layer is employed across the discretized samples to extract temporal features (width = 276) and the second across the electrodes to extract spatial features (width = 276). The temporal convolutional layer has a smaller kernel size (25 × 1) than the spatial convolutional layer (64 × 40), allowing for a larger range of transformations in this layer. The following mean pooling layer with a kernel size of 75 × 1 prevents overfitting. The final dense layer contains a 40 × 12 size feature map and outputs two predicted labels (correct or incorrect) using a logarithmic activation function.

#### 2.4.3. Siamese Neural Network

Based on the encouraging outcomes demonstrated in [13], we made the decision to explore the influence of the Short-Time (ST) estimate when employing a Siamese Neural Network. The architecture we constructed closely resembles the one outlined in the cited study, with a few customizations for our specific needs. These adaptation were necessary because the original study utilized the covariance matrix of the EEG signal as input for the classifier, whereas we directly used the EEG signal itself.

The Siamese Neural Network that we used consisted of two identical CNNs sharing the same parameters and hyperparameters. Each CNN is structured with two successive convolutional layers followed by two fully connected layers. We computed the Euclidean distance from the features extracted by these two CNNs, then employed this metric to determine whether or not the inputs to the two CNNs belonged to the same class. Training was carried out on all conceivable pairs of observations from the training set. To classify a new observation, we paired it with each observation from the training set. With the knowledge of the labels of the training data, we used a majority voting strategy to predict the class of new observations.

It is noteworthy that we made a deliberate choice not to apply any data augmentation techniques for this particular classifier. This decision was made in order to ensure that the comparison between pairs of observations remained exclusive to the original data, without the introduction of any synthetic data.

#### 2.4.4. Hyperparameter Optimization

To improve classification performance, a hyperparameters optimization process was performed. The pipeline was divided into four cycles, each one optimizing a different set of hyperparameters. Table 1 reports the different cycles and related sets of hyperparameters.

Each cycle included groups of hyperparameters related to one another; for instance, the hyperparameters involving the number of filters of the layers were included in the same testing cycle. The cycles were numbered to ensure that the first cycle would optimize the most important hyperparameters (i.e., the ones with the greatest affect on performances). The cycle was stopped when convergence was reached for a set of hyperparameters or after a high number of repetitions. Optimization was performed through *grid search* for those hyperparameters with a range known a priori; otherwise, a *random search* was performed. The metric used for optimization purposes was the *F1-score*.

Hyperparamater optimization was performed separately for each subject and for the population as a whole; thus, different sets of parameters were obtained for each analysis.

#### 2.4.5. Performance Metrics

The classification performance was evaluated using three different metrics: balanced accuracy, *F1-score*, and utility gain.

Balanced accuracy and *F1-score* were computed due to the dataset being highly unbalanced, which makes erroneous events much more rare than correct actions. When dealing with an unbalanced dataset, these metrics are preferable to accuracy, which is biased in such cases and does not represent the classifier’s real performance.

The utility gain is a metric introduced in [29] that represents the potential gain when introducing an error correction system in a BCI system. It is computed as follows:(12)$$g\left(p\right)=\frac{pr_{C}+\left(1-p\right)r_{E}+p-1}{2p-1},$$
where *p* is the performance of the BCI system with no correction, $r_{C}$ is the recall of correct events, and $r_{E}$ is the recall of erroneous events, with the latter two respectively computed using the following formulas:(13)$$r_{C}=\frac{TN}{TN+FP}$$
(14)$$r_{E}=\frac{TP}{TP+FN}$$

In particular, a utility gain higher than 1 indicates an improvement in BCI performance through the correction system, while a value lower than 1 indicates that it is counterproductive compared to the original BCI system.

## 3. Results

### 3.1. Classification Performance

This section reports the performance of the proposed two-stage procedure for classifying ErrP versus non-ErrP epochs. The results for the EEGNet, L-CNN, and Siamese Neural Network are presented separately in Section 3.1.1, Section 3.1.2, and Section 3.1.3, respectively. Our results are presented as a series of graph, each containing both subject-wise results (one for each subject) and population-based results (identified by the label *All Subjects*).

#### 3.1.1. EEGNet

The performance when using EEGNet to distinguish between ErrP and Non-ErrP classes is reported in Figure 2 in terms of the *F1-score* and balanced accuracy computed on the test set. The last group of bar plots, shown by the the black line on the right-hand side of each graph, is used to indicate the results obtained in the population-wise training.

In both the subject-wise and population-wise analysis, all the ST estimators led to an increase of performance compared to using raw data as input to the CNN classifier (the blue bar). This behaviour is observable for both the *F1-score* and balanced accuracy metrics. The largest increase in the metrics can be observed for subjects 4 and 6. For these two subjects, there are respective increases of 11% and 11.8% in terms of balanced accuracy and of 12.5% and 12% in terms of the *F1-score*. Notably, these two patients were had the two worst performances when feeding raw EEG data to the CNN. It is worth noting that the Subspace Regularization method resulted in the highest performance for all subjects (the orange lines in the figure).

The performance in terms of utility gain is reported in Figure 3, showing the improvement introduced by a correction system based on ErrP detection for a BCI with initial performance *p* in the range of 70–100%. In each graph, the curve above the others identifies the best-performing model. For all subjects and for the population-wise analysis (the bottom-right plot in Figure 3), most of the curves are above the blue line, demonstrating the utility of adding an ST-ErrP estimator stage. Among the ST methods, there was a slightly larger improvement when using the subspace regularization method (the orange curves). In particular, this method introduces a “productive” gain (i.e., g(p)>1) until a baseline performance of 90% is reached. It can be seen that this result is more prominent for subjects 4 and 6.

#### 3.1.2. L-CNN

Similar to the EEGNet results, we now report the results for the L-CNN classifier in Figure 4. An improvement in all metrics can be observed when the ST-ErrP extraction stage is performed. Similar to EEGNet, subspace regularization introduces the highest improvements for all metrics in most of the subjects as well as in the population-wise analysis. In particular, subjects 4 and 6 experience the highest gain in terms of performance, at 13.5% and 13% in terms of balanced accuracy and 12.8% and 12% in terms of *F1-score*, respectively. The main difference between these results and those obtained when using EEGNet is observed for subjects 1 and 5, where the CWT method method leads to the greatest improvements in performance.

The utility gain curves for the L-CNN classifier are shown in Figure 5; again, they show that the introduction of a correction system with subspace regularization for ST estimate improves the BCI performance in most cases, resulting in higher accuracy compared to the other techniques. This is particularly apparent for subjects 4 and 6, where a g(p)>1 is measured until p=84.7% and p=83.9%, respectively. For the other performance metrics, in terms of the utility gain, subjects 1 and 5 show more improvement when using the CWT to estimate the ST.

#### 3.1.3. Siamese Neural Network

Figure 6 reports the results for the Siamese Neural Network. In agreement with our previous results, domain-specific preprocessing improves the performances of ErrP detection for each subject as well as in the population-wise analysis. Greater improvements in both balanced accuracy and *F1-score* result when the subspace regularization method is used. For the other two architectures, subjects 4 and 6 experience the largest improvements in terms of performance, with 12.2% and 14% in terms of balanced accuracy and 12.6% and 13.4% in terms of *F1-score*, respectively.

The utility gain curves for the Siamese Neural Network classifier are shown in Figure 7. Again, it can be observed that the introduction of a correction system with subspace regularization for ST estimation leads to improved performance of the BCI and higher accuracy compared to the other techniques. This is particularly observable for subjects 4 and 6, where we measured a g(p)>1 until p=85.1% and p=84.7%, respectively.

## 4. Discussion

Our main findings in this research can be summarized as follows: first, the introduction of a stage dedicated to ST-ErrP estimation results in improved performance on the part of the CNN classifier on detecting the occurrence of ErrP; second, the method which provides the largest improvement is the one based on subspace regularization.

The first finding indicates that domain-specific processing may significantly reduce the complexity of the classification process, leading to improved performance of CNN classifiers when detecting epochs containing ErrP. The observation that this improvement is consistent for all three CNN models leads us to conclude that this is not a bias induced by the model itself, and may have general validity. While in principle the flexibility of CNNs might be supposed to easily incorporate the estimation of ErrP within the multi-layer perception network, our findings show that it is advantageous to relieve the CNN of this task by moving it from a minimal preprocessing task to a domain-specific one. It is not surprising that, when focusing on ErrP classification, the domain-specific preprocessing task can be performed effectively by an ST-ErrP estimator.

Among the investigated single-trial estimators, the subspace regularization method slightly outperforms the others in terms of improving the performance of CNN classifiers on the ErrP detection task. This method’s ability to highlight those EEG components related to error processing suggests its effectiveness in enhancing the quality of the recorded signal. In particular, it is worth noticing that this method is able to cut out the high-frequency components that are usually related to the background activity [30,31]. This filtering procedure results in epochs that closely resemble the expected ErrP waveform, which is a critical step in improving the detection of erroneous events [32].

Our evaluation of the utility gain indicates that the subspace regularization method not only permits allows effective classification of ErrP and non-ErrP events, it can enhance the overall utility of a BCI system as well. This finding suggests that the proposed architecture can provide significantly benefit in real-time BCI applications by enhancing error detection. It is noteworthy that subjects 4 and 6 showed the most improvement when using the subspace regularization method for extracting the ST. Indeed, these subjects initially had poor results when using raw signals as input for classification. This highlights the potential of the method described above to address inter-subject variability and provide improved performance for individuals for whom the model may initially struggle with ErrP detection.

In this study, we observed that when using the L-CNN classifier, the CWT resulted in better performance for subjects 1 and 5 than using the subspace regularization method. These subjects already showed good results when using the raw signals, and the improvement brought about by the two ST estimate methods were comparable. However, the differences in the results for these two subjects when using different CNNs suggest that the choice of classifier has an impact on the final signal processing performance. Further investigation into classifier-specific sensitivities could be beneficial, and might be addressed using Explainable Artificial Intelligence (XAI). From this same perspective, the adoption of an ErrP enhancer stage could facilitate the work of explainability approaches aiming to identify which parts of the signal contribute the most to classification. We hypothesize that the salient features highlighted by the ErrP enhancer are instrumental in simplifying the CNN classification task. Additionally, by reducing background EEG interference, this enhancement can contribute to the interpretability of XAI results and facilitate analysis of the relationship between relevant ErrP patterns used for classification and established evoked potential patterns. In future work, we intend to further investigate these aspects.

## 5. Conclusions

In conclusion, the present study provides compelling evidence that the proposed ErrP-enhanced architecture in which a CNN classifier is supported by a domain-specific preprocessing stage is highly effective in improving the accuracy of ST ErrP estimation and detection in EEG signals. The additional preprocessing stage offers several advantages, including the ability to filter out components that are not error-related and to enhance the fidelity of the ErrP waveform. These findings contribute to the ongoing advancement of EEG-based error detection systems, and have practical implications for BCI applications in domains such as assistive technology and neurofeedback.

## Figures and Tables

**Figure 1 sensors-23-09049-f001:**
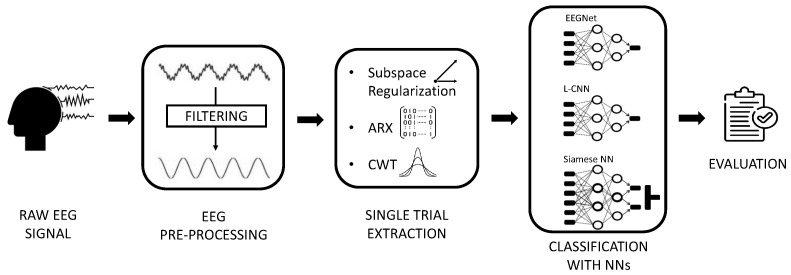
Pipeline of the employed methods. The novelty of our approach consists in the introduction of a Single Trial Extraction block between the EEG preprocessing stage and the NN classification stage.

**Figure 2 sensors-23-09049-f002:**
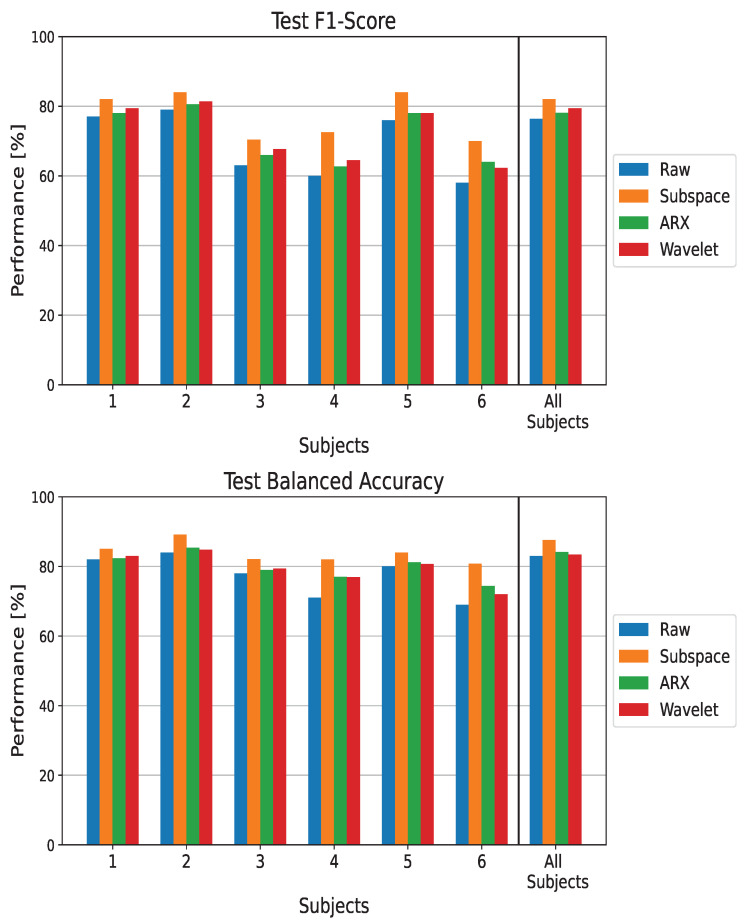
Performance metrics on the test set when using EEGNet as a classifier. The metrics are reported for each processing technique and in both subject-wise and population-wise analysis. The balanced accuracy and *F1-score* are reported for completeness.

**Figure 3 sensors-23-09049-f003:**
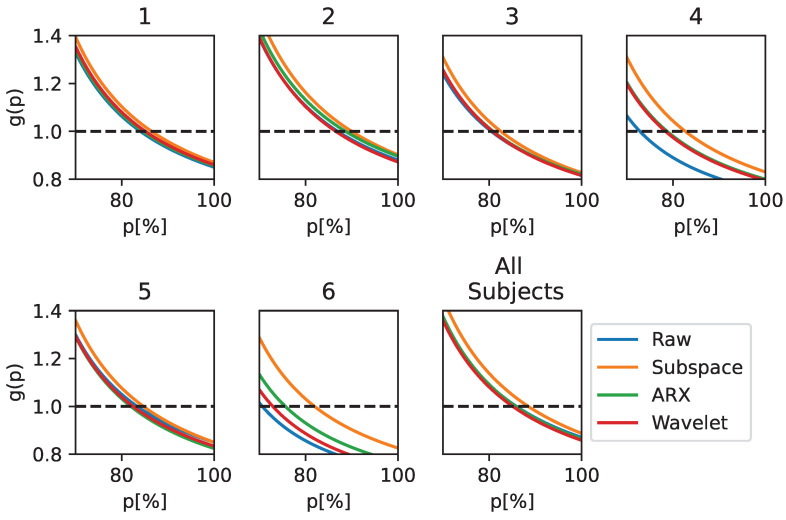
Utility Gain when using EEGNet as a classifier and applying different techniques for ST estimation. The metric is reported for the subject-wise and population-wise analysis for only *p* values over 70%, as no major differences were observed outside of this range.

**Figure 4 sensors-23-09049-f004:**
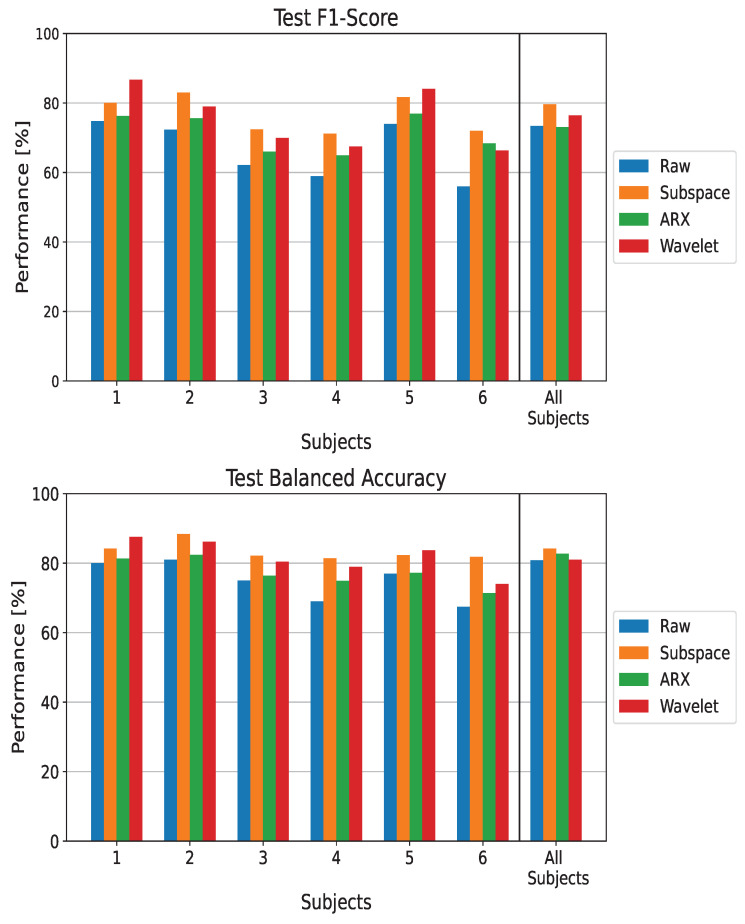
Performance metrics on the test set when using L-CNN as a classifier. The metrics are reported for each processing technique and in both the subject-wise and population-wise analysis. Balanced accuracy and *F1-score* are reported for completeness.

**Figure 5 sensors-23-09049-f005:**
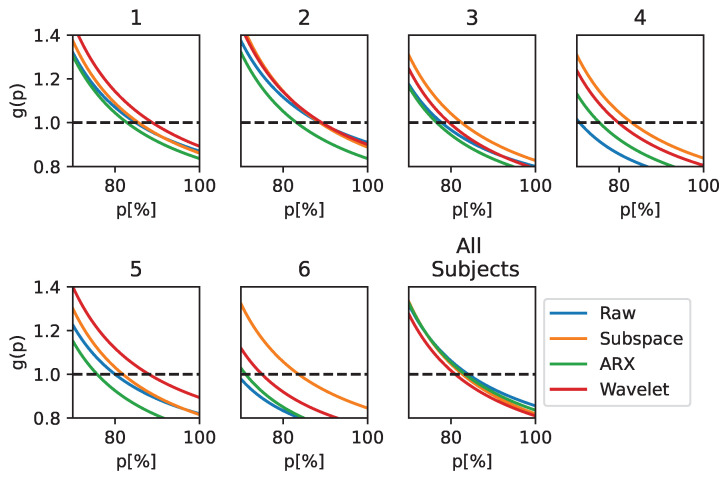
Utility gain obtained using L-CNN as a classifier and applying different ST estimation techniques. The metric is reported for subject-wise and population-wise analysis and for *p* values over 70%, as no major differences were observed below this level.

**Figure 6 sensors-23-09049-f006:**
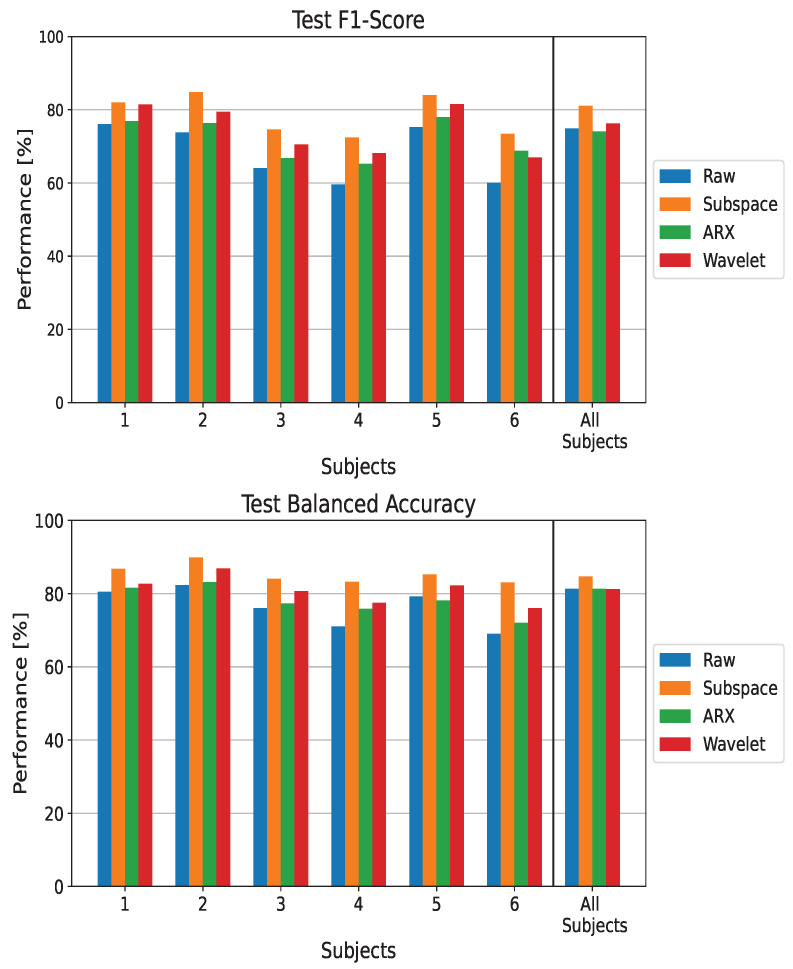
Performance metrics on the test set when using a Siamese Neural Network as the classifier. The metrics are reported for each processing technique and for subject-wise and population-wise analysis. Balanced accuracy and *F1-score* are reported for completeness.

**Figure 7 sensors-23-09049-f007:**
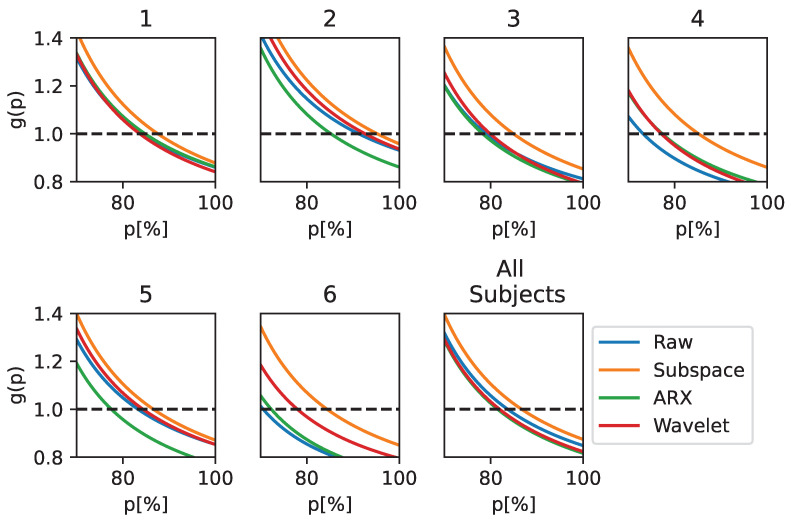
Utility gain obtained using a Siamese Neural Network as the classifier and applying different ST estimation techniques. The metric is reported for the subject-wise and population-wise analysis and for only those *p* values over 70%, as no major differences were observed below this level.

**Table 1 sensors-23-09049-t001:** Hyperparameter optimization: in each cycle, a different set of hyperparameters is optimized, with each set being composed of related hyperparameters. If convergence or a high number of iterations is reached, the cycle is considered to be optimized.

First Cycle	Second Cycle	Third Cycle	Fourth Cycle
learning rate	pooling layer		dropout rate
iterative method	dropout layer	conv layers size	momentum
batch size	activation layer	(F1, D, F2)	learning decay

## Data Availability

Publicly available datasets were analyzed in this study. This data can be found here: https://bnci-horizon-2020.eu/database/data-sets (013-2015).

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
