# Peer review of "Domain-Specific Processing Stage for Estimating Single-Trail Evoked Potential Improves CNN Performance in Detecting Error Potential"

_sensors, 2023, doi:10.3390/s23229049_

Round 1
Reviewer 1 Report
Comments and Suggestions for Authors
The present work presents a pretty standard procedure of applying neural network-based machine learning to classify the EEG single trials into two types (err and normal). While the overall presentation of the work is clear and sound, I am a bit afraid that the novelty of the work is a bit lacking.
Let me put it in this way, the same procedure can be applied to classify any EEG single trial data. It is a pretty prototypical routine.
The authors claimed that the main point is that the application of certain pro-processing/signal extraction methods lead to improved accuracy. This appears to be quite trivial too. Most of EEG pre-processing or signal extraction techniques are bearing an aim to extract more relevant information in terms of association with underlying cognitive factors (here, error processing). It is almost predictable that such signal extraction techniques will enhance the SNR or expose the relevant signal better.
Therefore, I would suggest that the authors try to organize the manuscript into a version that is able to identify a novel point to present. Otherwise, it reads more like a tutorial paper.
Comments on the Quality of English Languagelanguage needs to be substantially improved (lots of grammar issues/typos were spotted).
Reviewer 2 Report
Comments and Suggestions for Authors
In this paper, the authors introduce a two-stage architecture designed to improve the detection of Error Potential signals during ErrP stimulation tasks. The approach incorporates advanced Single-Trial ErrP enhancement techniques to process raw EEG signals in the initial stage, followed by the application of Convolutional Neural Networks to distinguish between ErrP and NonErrP segments in the second stage. Experimental results validate the model's effectiveness. However, there are several areas where the paper could be enhanced:
1. The abstract lacks a clear motivation regarding the research gaps that the two-stage architecture addresses, especially when compared to similar existing architectures.
2. The literature review in Section 1 needs better organization to highlight the research motivations. Consider creating a new section to explicitly state the contents and objectives of the literature review.
3. In Section 3, the comparative analysis should extend beyond classical methods to include recent advancements in neural networks. This broader comparison can provide a more comprehensive understanding of the model's performance.
4. Improve the quality of Figure 2 and provide a clear explanation of why the presented metrics were chosen to evaluate the model's performance.
5. It is essential to make the study's motives and contributions more evident in Section 1. Explain why the selected foundational tools were used and how they contribute to addressing the research objectives.
6. In the final section, consider summarizing the paper with insights into emerging topics in machine learning and explainable AI that relate to your research, providing a forward-looking perspective on potential directions for future work.
Comments on the Quality of English LanguageMinor editing of English language required.
Reviewer 3 Report
Comments and Suggestions for Authors
The submitted article makes significant contributions to the field of ErrP detection in brain-computer interfaces (BCI). Introducing a domain-specific processing stage focused on ST-ErrP estimation has notably improved CNN classifier performance in ErrP detection. This approach proves effective in reducing classification complexity and demonstrates consistent results across different CNN models. Additionally, the Subspace Regularization method stands out among single-trial estimators, enhancing ErrP detection and overall BCI system utility. The study also highlights its potential to address inter-subject variability, offering promising applications for individuals with varying proficiency levels. Moreover, it suggests exploring classifier-specific sensitivities using Explainable Artificial Intelligence as a future research avenue.
Round 2
Reviewer 1 Report
Comments and Suggestions for Authors
The ErrP enhancement block is an intermediate step that filters out more relevant information based on some prior knowledge about the features of ErrP. For me, this is a really trivial. There could be dozens of ways to enhance/filter the signal. I hope the authors at least argue why the three enhancement methods were selected. Otherwise, this work is really not contributing to the research community in enhancing neural signal detection.
Comments on the Quality of English LanguageLanguage needs to be substantially improved.
Reviewer 2 Report
Comments and Suggestions for Authors
The authors have addressed previous comments.
Round 3
Reviewer 1 Report
Comments and Suggestions for Authors
Non of the preprocessing (signal enhancement stage before CNN) is specific to ErrP. Actually, lots of parameters in the signal enhancement stages are determined without specific justifications. This is not a problem, though, if the methods aim to demonstrate that certain signal enhancement technique (e.g., some kind of filtering) is able to enhance signal thus raise the CNN accuracy. But those are not specific to ErrP. And as I mentioned in my earlier comments, those signal enhancement techniques are selected without a strong motivation (why are these selected from many others). Even in the signal enhancement method they selected, there are lots of parameters variations that can potentially lead to better accuracy outcome. Therefore, the main point in this work is that adding a signal enhancement step before feeding to CNN is recommended. I hope the manuscript can somehow organized in a way that truly reflect what the main contribution is, even when the main contribution may seem trivial.
Comments on the Quality of English LanguageEnglish MUST be improved. There is even typo in the title.